# From Research to Practice: Toward the Examination of Combined Interventions for Autism Spectrum Disorders

**DOI:** 10.3390/brainsci11081073

**Published:** 2021-08-16

**Authors:** Eric Bart London, Jiyeon Helen Yoo

**Affiliations:** Department of Psychology, New York State Institute for Basic Research in Developmental Disabilities, 1050 Forest Hill Road Staten Island, New York, NY 10314, USA; jhelenyoo@gmail.com

**Keywords:** combined intervention, psychopharmacology, antipsychotic medications, applied behavior analysis, health home, autism spectrum disorder, research designs

## Abstract

The use of biological (i.e., medications) in conjunction with applied behavior analysis is relatively common among people with ASD, yet research examining its benefit is scarce. This paper provides a brief overview of the existing literature on the combined interventions, including promising developments, and examines the existing barriers that hinder research in this area, including the heavy reliance on RCTs. Recommendations for possible solutions, including the creation of health homes, are provided in order to move toward a more integrated approach.

## 1. Introduction-to Combined Interventions for Autism Spectrum Disorders

Since Leo Kanner’s 1943 report describing eleven children with autism, considerable progress has been made in the treatment of autism spectrum disorders (ASD) with the application of applied behavior analysis and the use of antipsychotic medications. However, treatments that target the core symptoms remain elusive and the long-term prognosis for many individuals remains stagnant despite these advances [1,2,3]. Some treatments have substantial evidence of efficacy (i.e., ABA, risperidone, aripiprazole) while others have scanty evidence at best or lack evidence altogether. Parents, educators, and even physicians caring for people with ASD utilize proven but also untested therapies that range from promising to fraudulent and even outright dangerous. The vast majority of these available “therapies” are not supported by sufficient evidence as presently defined [4].

The core symptoms of ASD and the common comorbid conditions require a multifaceted treatment approach given that as many as 68% of children and adolescents with ASD exhibit co-occurring symptoms such as aggression, self-injury, hyperactivity, and rapid mood changes [5]. Currently, there is substantial evidence demonstrating the benefit of medication [6] as well as behavior interventions [7].

With the myriad of symptom presentations in ASD, the need for a personalized, multipronged approach should be obvious. The current research literature, however, reveals a startling lack of studies examining combined ABA (or other evidence-based educational and behavioral techniques) and medication (or other biologic techniques) [8]. Optimally, each therapy would selectively target undesirable symptoms but preserve and enhance the already compromised cognitive and adaptive skills and provide more than added benefit when combined. The lack of research and evidence of integrated combined treatment prevents the recommendations of combined treatment as a “best practice.” Recommendations for combined treatments are conspicuously absent in most clinical practice guidelines. When the absence of guidelines is combined with the realities of healthcare systems that rely heavily on such guidelines for service authorizations, there is a structural and monetary justification for overlooking the potential benefit of integrated treatment modalities. As a result of the paucity of research in this area, the existing clinical practice suffers as the available treatments exist as non-communicating silos within the medical system [9].

In a meta-analysis of 30 intervention studies of disruptive behaviors in intellectual and developmental disabilities (IDD) under which ASD falls, only eight studies utilized combination treatments. Of those eight, five studies combined psychotherapy with contextual intervention, two studies combined psychotherapy with biological intervention, and one study combined biological intervention with the contextual intervention [10]. The dismal number of studies using biological with other treatments is in contrast with the fact that 15–29% of people with IDD engage in chronic and severe behavior problems. In over 70 years since Kanner published his paper and despite two readily available evidence-based interventions, the integration and optimization of the two strategies have yet to be realized, even in the face of discouraging outcomes for many living with ASD and other IDDs.

The goal of this report is to briefly review the existing literature on combined biological and behavioral interventions, to show the feasibility of this type of study, to present studies that utilize interesting and progressive methodologies, to examine the existing barriers and provide a framework for possible solutions in maximizing the available and promising treatments for individuals with ASD. In doing so, we will review the heavy reliance on RCTs and the potential contributions of single-subject designs. We will suggest possibilities for creating medical or health homes that are a natural fit for combined biological and behavioral interventions. While we will focus on the lack of combined biological and behavioral treatment, our analysis could be relevant to other treatment modalities as well.

## 2. Combined Antipsychotic and Other Medications with Behavior Intervention

Two antipsychotic medications, risperidone and aripiprazole, are the only medications that are Food and Drug Administration (FDA) approved for the treatment of “irritability” (e.g., aggression, tantrums, and self-injury) associated with ASD. Therefore, the existing combined studies have been conducted with these two medications [11]. The first randomized controlled trial (RCT) examining the combination of treatments was conducted by the Research Units on Pediatric Psychopharmacology (RUPP). This study examined risperidone alone and in combination with behavior-analytic parent training in children with ASD [12]. Parent training alone was not examined. The children who were deemed non-responders to risperidone were switched over to aripiprazole. The results showed a significant improvement with combined treatment at 24 weeks (71% for combined treatment vs. 60% for medication alone). Children who received combination therapy required a lower dose of medication than children who received medication alone. A one-year follow-up showed that despite the gains observed in the combined treatment group, that benefit was diminished by more than half within a year. Furthermore, the lowered dose required in the combined treatment group was not maintained, indicating a need for continued monitoring and parent training [13].

The RUPP group later conducted a similar study examining the effects of combination treatment on adaptive behavior [11]. In this study, parent training was focused on managing serious challenging behaviors. Findings show a greater improvement in socialization and communication in the combined treatment group. However, this improvement was modest and not evident in all areas of adaptive functioning. The authors concluded that improvements noted in both groups suggest that a reduction in serious challenging behaviors may have led to improvement in adaptive behavior.

Risperidone alone and in combination with pivotal response treatment, a type of behavior-analytic therapy, was evaluated in young children with ASD [14]. Initially, there was no difference between the two groups but three months after treatment, the combined group showed a significant improvement in social withdrawal and in inappropriate speech. The authors speculated that the risperidone had no effect on learning and adaptive behaviors but did reduce behavioral problems, thereby readying the child to engage in communication and social interaction.

In a study of 32 youths receiving various antipsychotics or other medications for aggression, intensive behavioral interventions were added. Those on the antipsychotic medications allowed for significantly fewer sessions of the behavioral interventions to achieve the behavior plan success while this was not true for the mood stabilizers or non-stimulant medications [15].

Several studies have been published on atomoxetine and parent training for children with ASD by one group of investigators [16,17,18]. Attention Deficit symptoms in subjects with ASD were significantly improved with the medication alone, parent training alone, and combined treatment compared to placebo alone and the medication with or without parent training showed significant benefit for symptoms of non-compliance. At a 1.5 year follow up the outcome with parent training was not significantly better than treatment without parent training and most of the responders maintained their benefits despite discontinuation of the medication. In this case, the acute benefits of parent training alone were significant but the 12 sessions of parent training made little long-term difference.

In a study on ABA therapy with bumetanide for a three-month duration both treatments alone showed significant improvement from baseline and combined therapy was significantly superior to either treatment alone [19]. In a study of melatonin and cognitive behavioral therapy for sleep in ASD, it was found in a three-month study that the combined treatment was better than either arm alone, at least in the short term [20].

## 3. D-Cycloserine as an Adjunctive Agent; Methodologic Issues

There is literature suggesting that D-cycloserine might be beneficial in treating social skills deficits in ASD [21,22]. In a randomized, double-blind, placebo-controlled trial, D-cycloserine was used along with behavior-analytic social skills training for children with ASD. The children were randomized to receive either active D-cycloserine or placebo 30 min prior to the weekly group social skills training. The social skills training included behavior skills training that incorporated social stories in the presence of typically developing peers. Social skills improved in all children but D-cycloserine was no more effective than placebo [23]. The authors noted that the social skills training was very effective and might have created a ceiling effect.

The same group then followed up with the children at one week post-training and again 11 weeks later [24]. The D-cycloserine group maintained their benefit significantly better than those who took the placebo. Without delving into the issue of whether D-cycloserine is beneficial in ASD with or without social skills training, it is instructive to examine the research question and the methods of these studies as they could potentially serve as a template for future investigations of combination treatment studies.

Initially, the first study asked whether the combined treatment with D-cycloserine is superior to social skills training alone. In this case, the simple answer is no. The follow-up study, however, asked a more subtle question: Will the benefits of the combined treatment be extended compared to a singular treatment? Such a question is rarely asked in the literature and a long-term outcome study is even rarer to find compared to the customary 6–12-week clinical trials evaluating a treatment vs. placebo. This study may serve as a prototype for future investigations that seek to combine biological and behavioral interventions with long-term safety and an efficacy follow-up. The paradigm shift is that, as opposed to functioning as an antianxiety medication that directly treats the symptoms, D-cycloserine may serve as a facilitator of the learning process, enhancing the efficacy of the behavioral intervention. Since few of the medications tested in ASD have been tested in combination with behavioral intervention, the potential for this sort of mechanism remains unknown.

By examining the effects of D-cycloserine in other related disorders, the complexity inherent in studying treatment efficacy offers a cautionary tale around the premature interpretation of combined treatments. A single dose of D-cycloserine before exposure-based therapy enhanced the treatment outcome for acrophobia [25]. This was followed by studies on social anxiety, panic, post-traumatic stress disorder (PTSD), and obsessive-compulsive disorder (OCD), all showing that D-cycloserine significantly aided treatment outcomes when provided in conjunction with brief cognitive-behavioral therapy [26]. Taken together, is it possible that the role of D-cycloserine is to enhance the effectiveness of exposure-based treatments rather than as a standalone therapy? As more research accumulated, inconsistent findings were reported [27,28], including a study on PTSD in which the combined D-cycloserine and behavior therapy had a worse outcome than placebo combined with behavior therapy [29]. This led a meta-analysis to question whether D-cycloserine should be used at all as an augmentative agent for anxiety disorders [27]. At the same time, this meta-analysis also found a small but significant benefit of using combination treatment for social anxiety. Therefore, meta-analyses should be interpreted cautiously given that methodological rigors vary widely across studies. It is possible that when the research questions from each study are grouped for analysis, they may become over-simplified while the findings from a meta-analysis may be overgeneralized.

## 4. Neuromodulators; Promising Candidates for Combined Treatment

Neuromodulation is emerging as a potential clinical tool in treating ASD and related disorders [30]. Neuromodulators work on such parameters as neuronal excitability, set shifting, and arousal, among others. The effects of neuromodulation are thought to be optimized when a neural circuit is actively engaged by behavior intervention or rehabilitation during or in temporal relation to the biologically based treatment.

### 4.1. Cholinesterase Inhibitors

Four medications approved for Alzheimer’s disease (donepezil, galantamine, tacrine, and rivastigmine) are cholinesterase inhibitors, which work by preventing the breakdown of acetylcholine. In addition, galantamine also stimulates the nicotinic cholinergic receptors [31]. Cholinesterase inhibitors function as a neurotransmitter in the periphery, but centrally, they function as neuromodulators, changing neuronal excitability, altering the presynaptic release of neurotransmitters, and coordinating the firing of groups of neurons [32]

Two double-blind placebo-controlled studies of galantamine found significant improvements in both the core and associated symptoms of ASD [33,34]. The other medications showed inconsistent results, although not uniformly negative. All of the studies employed the traditional drug trial methodology, and none employed a combined treatment component. Given that medication can influence the efficacy of behavior therapy and nicotine has been shown pre-clinically and clinically to enhance sensitivity to reinforcement [35], a combined intervention based on nicotine and the learning principles (i.e., ABA) may be a promising area.

### 4.2. Noninvasive Brain Stimulation

Another neuromodulatory modality is using various types of energy as brain stimulators. Magnetic, direct and alternating current electricity, ultrasound, and microwaves are either in use or under development [36,37,38,39]. Along with transcranial magnetic stimulation [39], transcranial direct current stimulation (tDCS) has also been shown to have some preliminary benefit for ASD [40,41,42]. In the case of tDCS, there is strong evidence that it is effective in combination with behavior intervention or rehabilitation rather than as a monotherapy [43,44].

### 4.3. Noradrenergic Medications

The noradrenergic system regulates arousal, attention, learning, sensory processing, emotional regulation, autonomic regulation, and shifting in order to adjust to the changing environment [30]. Small sample-sized studies using single-dose noradrenergic medications have been shown to improve semantic fluid tasks, working memory, eye contact, and verbal problem solving [45]. Adrenergic agents have also been reported to reduce severe aggressive and self-injurious behaviors in people with ASD [45,46,47]. Most of the literature consists of case studies, although there have been several small double-blind, placebo-controlled studies involving patients with these symptoms [48]. These studies involved many different psychiatric diagnoses, including intermittent explosive disorder among people with various types of intellectual disabilities with various etiologic origins [49]. Definitive randomized double-blind placebo-controlled trials (RCTs) have not been conducted. Despite the urgent need for treatment for these debilitating symptoms, and the anecdotal success with noradrenergic agents for aggression across a range of diagnoses, it appears that it is not commonly used for ASD treatment. In systematic reviews of treatments for ASD, it is either not mentioned or is briefly mentioned in the context of there being no high-quality studies to demonstrate its efficacy [6,50]. For all of the above neuromodulators, we are not aware of studies using combined treatments.

## 5. Medical Home

A committee appointed by the National Academy of Sciences identified critical components of effective science-based educational practices for children with ASD but there was no mention of medical or biological interventions [51]. Although outside the scope of this committee, the exclusion of medical interventions may have inadvertently omitted the significance of combined interventions to fall outside of “best practices” [52]. This type of de facto separation of medical from non-medical services is not unique to ASD. It is the recognition of these silos that spurred the formulation of the medical home concept to enhance the coordination between diverse medical and allied specialties. The medical home concept was first introduced in 1967 as the standard for all primary care with a goal of providing comprehensive, coordinated, accessible, and family-centered care thereby improving the quality and reducing the cost of healthcare [53]. Despite its long history, medical homes for people with ASD remain largely aspirational [54]. Less than one in five children with ASD have a medical home and less than a third receive coordinated care [53,55]. As a result, children with ASD were four times more likely to have unmet health care needs than children with other special health care needs. Behavior therapy services was identified as the highest need [55]. Without the medical home for ASD, the separation and isolation of disciplines continue as a norm and is a reason for the lack of meaningful exposure to related disciplines. Structural and administrative obstacles, such as the lack of reimbursement for time spent consulting with other practitioners, make it very difficult to put into practice [56].

## 6. Evidence Based Practice

The goal of reviewing available intervention strategies is to define and elucidate supporting evidence to enable real-world adoption and implementation in clinical and educational settings. However, the Agency for Healthcare Research and Quality (AHRQ) noted the lack of evidence-based interventions in schools [57]. This discrepancy between expert panel guidelines and the adoption and application of such guidelines shows that evidence considered by the research community to be the “gold standard” is different from those embraced by educators and community organizations [58]. Similarly, the medical and behavioral models utilize very different research methodologies to assess the effectiveness of interventions. In 2002, a working group was convened by the NIMH to develop guidelines to help investigators and funding agencies evaluate psychosocial research in ASD. A separate committee reviewed psychopharmacologic agents [59]. The psychosocial research group recommended conducting efficacy studies using single-subject methodologies to identify promising interventions and to manualize those interventions. They then recommended conducting RCTs followed by community-based effectiveness studies. Again, the examination of combined interventions was excluded.

## 7. Randomized Controlled Trials (RCT)

In the medical model, the RCT is the gold standard for intervention research. It is ranked second only to the systematic reviews of the literature (e.g., Cochrane Reviews), which often only considers RCTs [60,61,62]. Two epistemological assumptions underlie the RCT. The first, associated with the statistician RA Fisher, maintains that randomization is necessary for the valid interpretation of statistical significance [63]. The second, associated with the epidemiologist Austin Bradford Hill, asserts that randomization prevents biased estimates of the value of new interventions. Hill is generally acknowledged as the first person to introduce the RCT into modern medicine [64]. It was during post-war England that his method was put into use because of an inadequate supply of expensive streptomycin for tuberculosis. Prior to this, most studies did not use the rigorous methods expected today [65]. Certainty was therefore needed for both medical and economic reasons. As for the limitations of RCTs, Hill himself pointed out that the results from positive RCTs only indicate that some of the subjects who participated benefited from the intervention but they “do not answer the practicing doctor’s question: what is the most likely outcome when this particular drug is given to a particular patient?” [66] and more sophisticated methods are needed to adequately address the individual patient’s course of treatment.

Despite such a caveat, there has been a growing belief in the supremacy of this method in the past half-century, with an unstated assertion that its supremacy is undisputable. The next major historical event was in 1962 when the United States Food and Drug Administration (FDA) mandated proof of efficacy with “well-controlled” studies, which has come to mean RCT exclusively. The RCT was first labeled the “gold standard” in a publication in 1982. The physicist Richard Feynman stated, “The statements of science are not of what is true and what is not true, but statements of what is known with different degrees of certainty” [67]. The concept of the RCT being the gold standard for preventive, diagnostic, and therapeutic interventions has been reviewed [68]. Major questions remain, such as, how much certainty we are deriving from the RCT and at what cost, compared to other methods with the clinician’s predicament in mind. How universally applicable are RCTs, especially for psychiatric and behavioral disorders? By what standard can the success of the RCT be measured, if it has been already crowned as the gold standard? The issue has a prominent role to play in understanding the current treatments of ASD and why medical and behavioral disciplines might be having so much difficulty in combining well-established evidence-based treatments.

In psychiatry, the most common criticism of the RCT is its limited generalizability to clinical practice [67,69]. Meta-analyses of the many psychiatric disorders using RCT frequently deliver contradictory messages. RCT may show improvement in scores of rating scales, however, these may not have ecological validity for patient outcome [69]. That is, a statistically significant outcome of an RCT may not yield a clinically meaningful improvement in patient symptoms. This has led to calls for effectiveness trials (as opposed to efficacy trials using the RCT methodology). Effectiveness trials use more “real life” parameters, such as broader patient selection. Due to the exacting nature of RCT and the large numbers of well-defined subjects needed to achieve statistical significance, the cost of these studies is formidable, leading to a scarcity of studies to answer the multitude of clinical questions.

As of July 2020, there were over a thousand studies registered with clinicaltrials.gov worldwide for ASD. Only four phase 3 and phase 4 RCT studies were actively recruiting and not sponsored by the pharmaceutical industry. Three of these were supported by the investigators’ institutions and one had foundation support. There were seven phase 2 studies with federal support. Clearly, gold-standard studies of ASD are few and far between. If RCT is our criteria, progress in discovering evidence-based treatments will be measured in decades, if not generations.

The biological treatments for symptoms of ASD have changed very little in the past 50 years and this is the case for much of psychiatry [70]. In using the gold standard of RCT and meta-analysis, the clinician is presently left with pitifully few options. In a review of 17 treatments in the Cochrane systematic reviews for ASD, the conclusion is that there was weak evidence for the benefits from acupuncture, gluten, and casein-free diets, early and intensive behavioral interventions, music therapy, parent-mediated early interventions, social skills groups, theory of mind cognitive modes, aripiprazole, risperidone, tricyclic antidepressants, and selective serotonin reuptake inhibitors (SSRIs) for adults. No benefits were found for sound therapies, chelating agents, hyperbaric oxygen therapy, omega-3, secretin, vitamin B6, magnesium, and SSRIs for children. What is termed weak evidence is “low to very low quality of evidence” [71]. Based on this as a standard, what treatment can we rightly call evidence-based? If clinicians were to accept this uncritically, it would likely lead to a situation in which the treatment of individuals with ASD would be very difficult or impossible. Clinicians need to weigh the risks and the benefits before prescribing a treatment. Considering the modest benefits if any (as per the RCT) vis-à-vis the potential side effects of medications, clinicians face a difficult choice. Compounding the problem is the increasing trend of managed care and prescription plans rejecting treatments that lack solid evidence. In such situations, if a physician’s clinical experience and acumen are not valued, the ability to treat this population which often presents with very serious and at times life-threatening behavioral symptoms would not be feasible. In a large epidemiologic study of medications used in five health systems, 48.5% of children ages 1–17 years with ASD were receiving psychotropic medications compared to 7% in those without ASD [72]. The authors prominently noted the irony of the widespread use of medications for ASD in the face of weak evidence. This sentiment is also commonly expressed by others, including parents, administrators, behavior analysts, and educators. Not surprisingly, such a state of the field leaves prescription of psychotropic medications a very unattractive and a doubtful alternative for many families.

These issues have been raised by investigators conducting treatment research in Fragile-X [73]. Fragile-X is phenotypically heterogeneous yet is a monogenic syndrome, which allows for a more coherent basic research based on cellular mechanisms. Despite good results in animal studies, large RCT studies with a GABA agonist and mGluR5 antagonists have failed. Considering several successful cases that were identified in post hoc analysis, the investigators noted that accurate stratification of the subjects in combination with the use of sensitive and specific dependent measures may have led to a different outcome. The RCTs have not been able to demonstrate improved outcomes for any urgently ill (ICU) patients despite the strong belief and the ongoing use of many treatments [74]. For example, in the case of palliative care, the nature of the conditions do not lend themselves to the RCT. Investigators have noted that in these and psychiatric disorders, other research methodologies are needed [75]. What these disorders also have in common is the heterogeneity of the etiologies and assortment of symptom presentations.

A similar predicament extends to psychosocial interventions. In a review of 50 years of psychoeducational interventions (i.e., ABA and cognitive-behavioral techniques) for severely affected adults with ASD, it was found that for the seven outcome domains (activities of daily living, aggressive/destructive behaviors, emotional functioning, language/communication skills, self-injurious behaviors, stereotypy/mannerisms, and vocational skills), there was reliable evidence only for improving emotional functioning. For all remaining domains, the support was rated as low or very low reliability [76]. The authors suggested, after acknowledging the RCT as the gold standard, expanding the range of future study designs beyond the RCTs.

## 8. Barriers to Overcome

This brings us to a critical question of whether these treatments with low-quality evidence “in reality” are beneficial. Lack of evidence is not equivalent to lack of benefit. One must instead seriously question whether the absence of evidence is a valid justification for inaction. Cynically, it could be argued that a substantial proportion of the roughly 50% of children with ASD who are taking psychotropics might be the result of a shared ignorance or delusion on the part of the prescribers and the children’s caregivers, continuing with the administration of ineffective medications despite potential side effects. This would entail both the physician and the caregivers not exercising reasonable judgment and thus, the allegedly ill-advised prescribing persists. To be sure this is the position of many. Alternatively, a placebo effect or other methodologic issues could make the discernment of medication effects problematic. If this is true, the scientific community has failed to appropriately modify the research methodologies to accurately evaluate and capture the genuine effects of the treatment.

Large pharmaceutical companies have been reducing their central nervous system (CNS) drug development programs due to a host of failed trials. The NIMH, which has grappled with this issue, announced that it would no longer entertain RCTs unless the submission is in response to a specific call for proposals [77]. This led to the Fast-Fail Trials (FAST) initiative program which requires a demonstration of biomarker effects prior to initiating the definitive phase 3 trials that are used for testing the compounds [78]. The ultimate success of this program remains to be seen; however, the FAST initiative is still based on the assumption that the RCT can definitively prove the impact of treatments that would then be used clinically. To restate the earlier question, is the RCT the best or even an adequate methodology to rely on for clinical outcome and subsequent guidance? Studying large groups of participants does not entirely solve the problem of generalizability of the findings to individual patients. The utility of the RCT for psychiatric (or other brain-based illness) may not be comparable to its value for other treatments such as antibiotics. The complexity of the CNS with the adult brain that comprises 86 billion neurons, an equal number of glia [79], 164 trillion synapses, and vast amounts of specializations cannot be anywhere as simple as finding the antibiotic for a single organism [80]. Nonetheless, the FAST initiative is banking on the prospective identification of biomarkers robust enough to lead to the creation of successful RCTs.

Another barrier is the validity of the ASD diagnosis itself. A diagnosis based on the Diagnostic and Statistical Manual of Mental Disorders (DSM) or International Classification of Diseases (ICD) is generally required as an inclusion criterion in RCTs despite the characterization of ASD as a highly uncertain entity without clear biological and behavioral demarcations and failed attempts to externally validate ASD [81]. In addition, the DSM and ICD diagnoses have been described as poor mirrors of nature [82,83,84]. Categorical diagnoses indicate little about etiology, lack biomarkers, have variable and poor predictive prognoses, frequently present with comorbidities, and are not predictive of treatment response. They also merge into other disorders and into neurotypicality [85]. A more pernicious element has been the reification of these diagnoses—making something concrete in the absence of evidence—thus similar to the RCT, diagnostic categories have taken on an axiomatic quality not to be challenged [84]. The weight and the prominence of the categorical diagnosis exist even though there are almost as many hypothesized causes of ASD as there are cases. As a result, serious discussions have occurred around whether ASD should be considered a valid diagnosis or perhaps it would be better off eliminated [83,86,87]. Reflecting from a sociological perspective, [81] one study queried individual researchers as to how they conceptualize their work, taking into account their doubts concerning the ASD diagnosis, which they all admittingly shared. Despite variable approaches of controlling for such uncertainty, all agreed that the diagnostic weakness is a problem.

Using the “gold standard” assessment instruments (e.g., ADOS-2, ADI-R) for the verification of ASD has allowed for improved diagnostic precision but of a diagnosis that may not be a natural kind. Given that many disorders have numerous overlaps in both observable behaviors as well as molecular findings, uncertainty remains [88,89,90,91,92,93]. Another NIMH initiative, the RDOC program, addressed this concern by requiring investigators to identify a circuit-based pathology without factoring in categorical diagnosis [94]. While this may be an advantageous and enduring precedent, categorical diagnosis is unavoidable when pursuing research through other funding mechanisms for the foreseeable future.

Clinically, the diagnostic ambiguities also lead to problems. The developmental disorders are generally not comprehensively treated by solo practitioners. Those with a diagnosis are referred to providers who specialize in some, but not all, facets of ASD. This problem has prompted the development of the concept of the Early Symptomatic Syndromes Eliciting Neurodevelopmental Clinical Examinations or ESSENCE [95]. Here, the treatment target is the individual symptom rather than the simple lumping of ASD into one set of treatments. Those symptoms, which are termed a comorbid condition may or may not be available in an ASD clinic. This can lead to missed opportunities to treat problems in a timely fashion if they are not routinely screened and recognized by the individual provider’s standard procedure.

The medical model, in most instances, prescribes treatment to restore the previous functioning and a successful treatment restores functioning back to baseline. In ASD, especially in young children, the goal is to develop skills that are delayed or absent. For example, if a child with no spoken words learns to say 100 words in a year, is that a successful treatment outcome? In the same time period, his neurotypical peers may have acquired over 1000 words. Standardized rating instruments can only provide a rough estimation of success, which is an elusive concept. Each child with ASD will start at a different baseline and progress at a different rate with or without treatment. The standardization of outcome measures is therefore very problematic for this reason and adopting and modifying instruments from other related disorders may not offer the sensitivity and specificity required especially when doing group studies.

## 9. Naturalistic Observations

Nearly all the classes of psychopharmacologic medications in use today were discovered between 1950 and 1970. Donald Klein, an early pioneer in psychopharmacology, recalled this time as disorienting, deliriously exhilarating, and enchanting [96], which are all rather extraordinary words coming from a scientist. The medications discovered during this time include lithium, lysergic acid diethylamide, chlorpromazine, iproniazid, reserpine, imipramine, chlordiazepoxide, haloperidol, and clozapine. Since then, there has been a 50-year lull with relatively little progress. Klein noted that many of the early discoveries were made serendipitously. The key to meaningful progress was careful clinical observation. Patients taking medications for various, often non-psychiatric reasons were noted to improve. Studies were often done in long-term state hospitals having much time for unstructured observation. In contrast, today’s health care system has markedly reduced doctor-patient time. This change in method is also in parallel with the neuroscience revolution, in which rational discovery replaced naturalistic observation and serendipitous discoveries. Klein himself acknowledged the prospect of duplicating the observational methods of the mid-20^th^ century is unlikely. As a response to a lack of progress, the NIMH has chosen to double down on its neuroscience base. Only time will tell if this is the right strategy.

## 10. Single Case Research Designs

While the medical community has abandoned the observation-based model described above, behavioral psychology and education, and in particular, the behavior-analytic community has continued to utilize and refine the methodology closer to that described by Klein. Behavior analysts rely on single-subject quantitative research designs and methodologies because behavior is deemed a result of an interactive relationship between the individual and the environment and it is, therefore, a phenomenon that occurs at the level of the individual [97]. Much less emphasis is made on comparison groups and statistical significance [98]. Single-subject designs are suitable for intensively studying and testing the effectiveness of treatments where the goal is to achieve clinically and socially meaningful change. The emphasis is on specifically targeted, operationalized behaviors rather than etiology, diagnostic entities, or early developmental history [99]. In addition, behavior analytic research is inductive rather than hypothetico-deductive, which avoids making large inferential leaps based on theories with limited data [100].

The single-subject designs, much like the RCT, have advantages and disadvantages. The disadvantage of single-subject designs is the method of data analysis, which relies on visual inspection of the dependent variable rather than statistical analysis. Visual inspection may be not sensitive enough to detect weak effects and can be unreliable, with different researchers reaching different conclusions about the same set of data. In addition, the results of visual inspection cannot be compared across studies unlike the measures of correlational relationships typically used in group research (Danov and Symons, 2008).

Single-subject research examines causal relationships and therefore offers good internal validity. Repeated measures of the same behaviors during baseline and other phases of the study offer assurance that threats to internal validity (i.e., maturation, instrumentation, statistical regression, testing) are controlled. In addition, single case studies can be conducted in the subject’s natural environment, and then the findings can be generalized across the subject’s behaviors and other people and situations [101]. This deficit in skill generalization is common in people with ASD. It is believed to reflect the neurologic deficit in which only specific associations are made rather than having the ability to make more general associations and integrate them with previous knowledge sometimes termed “priors” [102].

Despite the advantage of customizing treatments based on each person’s response, external validity is compromised in single-subject designs. That is, the applicability of the treatment outcomes to other individuals is unknown. This is a severe limitation of the single-subject study. While the RCT strives to define inclusion criteria, in single case studies, the subject is, by definition, unique and the results only apply to that individual. The heterogeneity of ASD makes this lack of generalizability a liability. On the other hand, large group studies do not entirely solve the problem of generalizing the findings to other individuals. Single-subject research can, however, point to the fact that the principles of classical and operant conditioning were discovered using the single-subject methodology and the results have been successfully replicated and generalized across a wide range of subjects and situations.

Currently, these single-subject research designs are deemed quasi-experimental and given less weight or are dismissed entirely for failing to meet the rigors of the medical model. The behavior-analytic attention to the environmental contingencies is not conducive to RCTs due to the challenges of controlling for the environmental variables in a given group. The RCT is more definitive and valuable for narrow and well-defined questions but a poorer method of studying more complex problems with many variables. Its optimal role is to demonstrate certainty once the medical, psychological, and scientific issues are exposed and well understood. We are a long way from achieving this level of proficiency but despite such shortcomings, RCTs are undertaken without a thorough understanding of the effects to be truly tested and the variables that need to be controlled [73,103].

Even after countless studies conducted since the 1960s, the field of ABA continues to strive to prove that its interventions developed using single-subject research designs are “evidence-based” because of the dearth of rigorous, high-quality RCTs within the field [104]. The assumptions and methods of ABA and the heterogeneity of ASD in combination with the ethics of conducting RCTs in this population hinder the demonstration of the efficacy of ABA to the extent needed to satisfy the medical model [105,106,107].

## 11. Potential Solutions

A prerequisite to agreeing on how to evaluate combined treatments is a resolution of the value of these different methods of research. At the very least, single-subject research and RCT should be considered complementary methods, with different strengths and weaknesses that are appropriate for answering different kinds of research questions. To bridge the methodological gap and to facilitate the integration of combined treatment, deliberate steps must be taken to achieve the following: (1) Physical and conceptual liaison between the disciplines; (2) agreements on the methods of defining and delineating “evidence-based” treatment; and (3) cooperation and recognition of the complexities of treatment research on the part of the many stakeholders, including families, administrators, insurance companies, government regulators, and clinical and research funding agencies.

Although there has not been a success in creating unified programs, one potential model could be the creation of a type of Health Home in an educational setting [9]. At the present, pediatricians and psychiatrists have little interface with the school services [108]. The educational setting is far and away the best place to treat young children and adolescents with ASD. Education is universally mandated, and students spend the majority of their day there during their most crucial developmental years. The student’s school (i.e., educational home) is a performance-based environment where social behavior and cognitive challenges are most apparent. Students begin spending a bulk of their time receiving special education and related services in this setting once the child reaches elementary school age [109]. Despite the practicality and efficiency of delivering treatment in the schools, it has been noted that university-sponsored, evidence-based intervention may not be sustained in schools or it may be modified to such an extent that it does not resemble the original intervention [58]. In a survey of teachers of students with ASD, fewer than 10% of the strategies used with the students were based upon scientifically based practices [110]. The gap between what is considered evidence-based educational interventions and the educational system is wide.

The disparity between the educational system and biological interventions is even wider. While about half of children with ASD are prescribed medications to treat challenging behavior [72], the interventions and support services often remain fragmented across the siloed systems of care [111]. In the United States the Individuals with Disabilities Education Act and Section 504 of the Rehabilitation Act of 1973 require schools to administer, monitor, and support students who are prescribed psychotropic medications. However, only 11 states were found to have policies in place to address psychotropic medications in the schools. Of those, the majority only broached the topic in order to prevent staff members from recommending these medications rather than supporting the optimal use of medications [112]. Little attention is given to training staff on how to best administer and properly monitor medications. Although school psychologists are the most highly trained staff to work with these students, only 20% of them receive any coursework in child psychopharmacology [113].

Despite the federal regulations, the reality is that educators and school personnel are not trained and even discouraged from having any role in supporting students who take medication. Communication between the teacher and the prescriber is rare [111]. In a survey of teachers working with students with ASD, more than half were unaware that their student was prescribed medication and of those who were aware, none reported being conferred with regarding behavioral observations or side effects [114]. Ostensibly, the providers were focused on the use of medications for observable behavioral symptoms. As is, the currently constituted system is nowhere near ready to make the sophisticated discriminations required to evaluate medication effects, let alone the combined medication and behavioral interventions on learning and behavior.

There are existing models that can serve as a prototype, requiring only small modifications for people with ASD. One such model is the School-Based Health Center [115]. In this model, primary care and specialty services such as dental care, mental health care, and case management are physically available within the school building. For ASD, additional disciplines, including behavior analysis and child psychiatry, could be added. This model was originally developed for the treatment of HIV has since expanded to include substance abuse prevention. Many, if not all, of the above issues, could be addressed with this administrative arrangement if fully supported.

## 12. Integration of Clinical and Research Silos

Any progress towards well-integrated biological and behavioral treatments will require major changes in the way “business as usual” is done on many levels. The existing silos of treatment and research must be dismantled. A first step would be to produce research demonstrating the benefits of combined treatment. While this has been done (albeit rarely) [11,14,116], the occasional study will not move the field along. Each study needs to ask a very specific question. In addition, research on combined treatments must be demonstrated broadly and replicated in order for clinicians to modify their practice. Currently, a growing number of stakeholders are represented in the examination room. Insurance companies, administrators, supervisors, attorneys, media, government regulators, quality assurance committees, human rights committees, and even the internet are all present. Qualified or not, all will have a voice in deciding the treatments being prescribed and administered. Well-meaning expert panels offer “best practices” [59], however, the term “best practices” can only mean current best practices based on the literature. Unless deliberately approached in a meaningful manner, it could take a very long time for combined treatments to emerge as best practices. Despite the great advancements in neuroscience and clinical medicine, progress for ASD will continue to be measured in decades, half centuries, or perhaps more.

To facilitate research on combined treatment, the appropriateness of a single research method—the RCT—has to be challenged [62]. There are many ways to demonstrate evidence. Those who have questioned the gold standard RCT have not recommended a free for all, but to consider various methodologies of documenting efficacy [51,104]. The reality of treatment decisions is that they are made with evidence at hand, which includes studies in the literature but also from one’s own clinical experience. Insistence on a system that is binary—either there is incontrovertible evidence or there is not—is marginally useful to the clinician in the real world. Other research methodologies need to be taught, encouraged, used, and funded. This can only be accomplished through the efforts of the funding agencies themselves. On a deeper level, funding and regulatory agencies take their cues from the academic community to make their decisions. It is the responsibility of the academic communities to acknowledge the problems outlined above. There is no reasonable argument for combined treatment research to be brushed aside. It is up to the academic community to advocate for this research to proceed.

Likewise, the various disciplines must learn to communicate outside of their disciplines. Teachers, behavior analysts, and other treatment providers need to receive some basic training in biologically-based interventions. Similarly, physicians must be aware of the non-medical evidence-based modalities. The American Psychological Association, as early as 1993 has formed committees to study the issues of disseminating research findings as evidence-based.

The above-listed suggestions are a top-down pathway. A bottom-up pathway would be promoting and supporting programs that combine evidence-based biological and behavioral treatments. Demonstration programs, specifically the evaluation of combined treatments, could be created and likely would lead to improved efficacy and cost reductions. This is the concept underlying the medical home. If there was tangible support to make the necessary adjustments to the medical or health home model for people with ASD, a variety of treatment options, including combined treatment, could follow more naturalistically. ASD can be a devastating disorder with poor outcomes for many. However, much could be improved with thoughtful adjustments to the existing systems in order to maximize our efficiency in studying and treating this condition.

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
