# Peer review of "From Research to Practice: Toward the Examination of Combined Interventions for Autism Spectrum Disorders"

_brainsci, 2021, doi:10.3390/brainsci11081073_

Round 1

Reviewer 1 Report

Brain Science manuscript 1310074 provides a brief literature review of behavioral-pharmacological interventions for people with autism spectrum disorder (ASD).  Research on psychotropic medications to treat behavioral challenges in people with ASD or related intellectual disabilities dates back to the 1970s.  The most cited literature on this topic is from research by Michael Aman, Nirbay Singh, and Stephen Schroeder.  It is a relatively well developed literature with much of the research funded by the NIH (i.e., NICHD, NIMH).  Unfortunately, it appears as if none of this research was incorporated into the manuscript under review.  This is regrettable because much of the work on risperidone and other antipsychotics was therefore not cited or reviewed in this manuscript.  Overall, the manuscript suffers from selective citation and a limited understanding of the existing pharmaco-behavioral literature.

The manuscript would also be improved if there was a clearer focus on population definitions and study inclusion criteria (e.g., ASD is a broad spectrum and treating Rett’s Syndrome and high-functioning autism as a single category requires an explicit rationale).  Also, there is no clear delineation regarding what “behaviors” were selected for review and why (e.g., in one section the focus is on challenging behavior and in another section the focus shifts to social competence, but no rationale is provided).  Similarly, the selection of pharmacotherapies is similarly underdeveloped with no clear rationale for why specific medications were selected for review and others excluded.   

Overall, the manuscript lacks a cohesive and clear thesis for what literature is being reviewed and the scholarship evident in the paper is lacking in merit.  I regret writing such a negative review, but the authors need to more carefully review this literature and provide a more compelling treatment of the existing literature for the paper to make a contribution to the existing literature. 

Author Response

Response to reviewer #1

Reviewer #1 Brain Science manuscript 1310074 provides a brief literature review of behavioral-pharmacological interventions for people with autism spectrum disorder (ASD).  Research on psychotropic medications to treat behavioral challenges in people with ASD or related intellectual disabilities dates back to the 1970s.  The most cited literature on this topic is from research by Michael Aman, Nirbay Singh, and Stephen Schroeder.  It is a relatively well- developed literature with much of the research funded by the NIH (i.e., NICHD, NIMH).  Unfortunately, it appears as if none of this research was incorporated into the manuscript under review.  This is regrettable because much of the work on risperidone and other antipsychotics was therefore not cited or reviewed in this manuscript.  Overall, the manuscript suffers from selective citation and a limited understanding of the existing pharmaco-behavioral literature.

Response- There seems to be a fundamental misunderstanding of the goals of this paper (although I think we are clear) “research on psychotropic medications to treat behavioral challenges”  as the reviewer puts it, is not the focus of our paper. We are focusing on combined treatments and more specifically the structural and methodologic difficulties in doing this type of research and treatment. We define combined treatment as applied behavior analysis (ABA) or similar behavioral treatments combined with a biological treatment. Further we had no intention of making this a review of specific treatments. We did discuss the studies of antipsychotic medications risperidone and aripiprazole with parent training or pivotal response training as an example of combined treatment. Contrary to the opinion of the reviewer, we contend that combined treatment does not have a well- developed literature, which is in fact the point of the paper. We site the one review of the topic we found, Heyvaert, Maes and Onghena 2010. Which found only 3 out of 30 studies of disruptive behaviors in Intellectual Developmental Disabilities (which most often will be patients with ASD) used  combined biologic with non biologic  treatments. Another review done in 2009, by Weeden et al laments the lack of combined risperidone and ABA studies and suggests that this be done.  

 After doing an extensive search (including all the works of the three investigators cited by the reviewer we found only one major study of combined treatment which we did not report on, and that was atomoxetine with parent training. We have described and added references to that study. We found three other studies using combined treatments one using bumetanide and ABA , in which they compared ABA alone to ABA and bumetanide (with no bumetanide alone arm) , a study using melatonin combined with cognitive behavioral therapy, and one study which added ABA treatment to aggressive youths with ASD who were already on various medications. We have added mentions of these studies to our submission. The addition of these studies does not change our overriding hypothesis that combined treatments remain rare, are difficult to do and methodologic concepts are in need of serious review.

We are not sure what the reviewer means by selective citation. If the implication is that we did not report on studies which contradicted our contention that combined studies are rare, important and difficult to do, we reject that assertion..

Reviewer #1 The manuscript would also be improved if there was a clearer focus on population definitions and study inclusion criteria (e.g., ASD is a broad spectrum and treating Rett’s Syndrome and high-functioning autism as a single category requires an explicit rationale).  Also, there is no clear delineation regarding what “behaviors” were selected for review and why (e.g., in one section the focus is on challenging behavior and in another section the focus shifts to social competence, but no rationale is provided).  Similarly, the selection of pharmacotherapies is similarly underdeveloped with no clear rationale for why specific medications were selected for review and others excluded.   

Response- Again the reviewers fundamental misunderstanding of the aim of this paper seems to be a problem. Our primary interest is in the research methodology  and clinical infrastructure issues, which make studies of combined treatments difficult and rare, and therefore not labeled as best practices,   It would be beyond the scope of our paper to review subcategories of ASD, the  symptoms treated or the specific medications which have been used in studies as the methodology issues and the infrastructure issues are generally the same. We agree with Reviewer #1 that the heterogeneity of ASD is a major concern. I have written about this topic previously in articles focused on that topic,

London, E. B. (2014). Categorical diagnosis: a fatal flaw for autism research?. Trends in neurosciences37(12), 683-686.

 Waterhouse, L., London, E., & Gillberg, C. (2016). ASD validity. Review Journal of Autism and Developmental Disorders3(4), 302-329.

and we discuss this issue in the text at fairly great length.

Reviewer #1

Overall, the manuscript lacks a cohesive and clear thesis for what literature is being reviewed and the scholarship evident in the paper is lacking in merit.  I regret writing such a negative review, but the authors need to more carefully review this literature and provide a more compelling treatment of the existing literature for the paper to make a contribution to the existing literature. 

Response-

This is taken from our paper lines 69-73

The goal of this report is to briefly review the existing literature on the combined biological and behavioral interventions, including promising developments, examine the existing barriers and provide a framework for possible solutions in maximizing the available and promising treatments for individuals with ASD. In doing so, we will review the heavy reliance on RCTs and the potential contributions of single-subject designs. We will suggest possibilities for creating medical or health homes that are natural fit for combined biological and behavioral interventions. While will focus on the lack of combined biological and behavioral treatment, our analysis could be relevant to other treatment modalities as well.

I think we do have a cohesive and clear understanding of the goals for this paper.

I also find it bizarre that this reviewer focused only on a review of specific treatments. In our submsission,  this was completed by line 106, There is no mention of any of the content of our submission from lines 107 to 647 obviously the bulk of the paper.

In summery we are adding mentions of the studies on atomoxetine, bumetanide, melatonin and the one study using various medications and ABA We hope this will be sufficient.  We are not interested in nor does the literature call for a review of the treatments of ASD.

Reviewer 2 Report

This was a very interesting and timely article as it emphasized that our current practices for evaluating the effectiveness of treatments with individuals with ASD and IDD need to be re-evaluated and that single-subject design may offer the individualization needed to assess significance of change given the heterogeneous nature of ASD.  I would refer the authors to the attached document - which states some criteria for defining empirically validated. It is an older document but might provide support for single-subject design more broadly. I would recommend publication with minor edits as there are some typos throughout and references in text that are missing from the reference section. 

Author Response

Reviewer #2

This was a very interesting and timely article as it emphasized that our current practices for evaluating the effectiveness of treatments with individuals with ASD and IDD need to be re-evaluated and that single-subject design may offer the individualization needed to assess significance of change given the heterogeneous nature of ASD.  I would refer the authors to the attached document - which states some criteria for defining empirically validated. It is an older document but might provide support for single-subject design more broadly. I would recommend publication with minor edits as there are some typos throughout and references in text that are missing from the reference section. 

Response to reviewer #2

We thank reviewer #2 for his kind words.

We have reviewed the document suggested  and have incorporated its content into the paper.

We reviewed the paper for typos and improved the  language of the submission and made corrections and have reviewed and corrected our reference section.   

Round 2

Reviewer 1 Report

The authors have adequately addressed my concerns from the previous review and I recommend publication of the paper.